# Carbon Nanomaterials (CNMs) in Cancer Therapy: A Database of CNM-Based Nanocarrier Systems

**DOI:** 10.3390/pharmaceutics15051545

**Published:** 2023-05-19

**Authors:** Hugh Mohan, Andrew Fagan, Silvia Giordani

**Affiliations:** School of Chemical Sciences, Dublin City University, Glasnevin, D09 NA55 Dublin, Ireland; hugh.mohan3@mail.dcu.ie (H.M.); andrew.fagan@umail.ucc.ie (A.F.)

**Keywords:** carbon nanomaterial, chemotherapy, nanocarrier, carbon nanotube, graphene, theragnostic, targeted drug delivery, formulation, carbon dot, side effects

## Abstract

Carbon nanomaterials (CNMs) are an incredibly versatile class of materials that can be used as scaffolds to construct anticancer nanocarrier systems. The ease of chemical functionalisation, biocompatibility, and intrinsic therapeutic capabilities of many of these nanoparticles can be leveraged to design effective anticancer systems. This article is the first comprehensive review of CNM-based nanocarrier systems that incorporate approved chemotherapy drugs, and many different types of CNMs and chemotherapy agents are discussed. Almost 200 examples of these nanocarrier systems have been analysed and compiled into a database. The entries are organised by anticancer drug type, and the composition, drug loading/release metrics, and experimental results from these systems have been compiled. Our analysis reveals graphene, and particularly graphene oxide (GO), as the most frequently employed CNM, with carbon nanotubes and carbon dots following in popularity. Moreover, the database encompasses various chemotherapeutic agents, with antimicrotubule agents being the most common payload due to their compatibility with CNM surfaces. The benefits of the identified systems are discussed, and the factors affecting their efficacy are detailed.

## 1. Introduction

While cancer remains one of the world’s leading causes of death, advances in diagnostics and treatment have seen an overall improvement in detection and mortality rates. However, the current treatment approaches are either highly invasive, in the case of surgical operations, or can cause unwanted toxic side effects, as commonly experienced with chemotherapeutic agents and radiotherapy [1,2]. In particular, the effectiveness of chemotherapeutic agents is often limited by their poor aqueous solubility and nonselective nature, resulting in poor bioavailability and the indiscriminate death of both healthy and cancer cells [3]. To overcome these issues, there has been much research into the use of nanocarriers for the targeted and controlled release of anticancer drugs [4], where carbon nanomaterials (CNMs) have emerged in recent years as very promising candidates for this purpose. CNMs are a distinct class of materials that show altered characteristics to those of bulk carbon materials, such as diamond or graphite. They are classified as 0D, 1D, or 2D, according to the number of dimensions they possess which exist on the nanoscale (<100 nm) [5]. The allotropic nature of carbon means that a variety of these materials exists, some examples of which include graphene [1], carbon nanotubes (CNTs) [6], carbon nano-onions (CNOs) [7], nanodiamonds (NDs) [8], and carbon nanohorns [9]. CNMs have garnered widespread attention for their biomedical applications, such as drug delivery and diagnostics, because of their unique and highly desirable physicochemical and mechanical properties, such as size, biocompatibility, high tensile strength, and ease of chemical functionalisation. 

By carefully selecting the production method, the particle sizes of CNMs can be precisely controlled, allowing for the creation of particles comparable in size to biomolecules (<100 nm) [10]. This size control enables CNM-based particles to take advantage of the leaky vasculature surrounding tumour cells through the enhanced permeation and retention (EPR) effect, facilitating the passive targeting of tumour cells [11]. However, passive targeting is generally limited, as not all tumours exhibit the EPR effect, and the random nature of the process makes it difficult to control and can lead to drug resistance [4]. Instead, it is preferable to actively target tumour tissues using targeting moieties that improve drug uptake through mechanisms such as receptor-mediated endocytosis [12]. The advanced surface chemistries provided by CNMs enable the attachment of various targeting ligands (such as folic acid (FA) [13]), imaging agents (such as BODIPY [14]), and anticancer drug molecules (such as cisplatin [15]), facilitating the creation of multifunctional nanocarriers. These nanocarriers can efficiently target, image, and deliver therapeutic agents directly to cancer cells, capitalizing on the unique properties of CNMs. Therefore, CNMs can be used as scaffolds to create theragnostic systems, combining imaging, detection, and treatment modalities in one tiny package to effectively diagnose and treat various illnesses [16]. 

As mentioned previously, there are many options for functionalising CNMs, with oxidation being one of the most straightforward approaches. This method introduces hydroxyl, carbonyl, and carboxyl groups to the surface of the nanomaterial, allowing for further functionalisation, and significantly increasing the material’s aqueous solubility in the process. Highly soluble CNMs can be utilised to increase the solubility of hydrophobic drugs, an approach taken by Cakmak and Eroglu, who employed graphene oxide (GO) to solubilise tamoxifen [16]. Facilitating the delivery of poorly soluble and/or poorly permeable drugs is a major benefit of the nanocarrier approach, as it does not require extensive modification of the drug molecule itself.

A range of other examples of covalent CNM modification exists, such as amidation, fluorination, and alkylation [17]. Covalent functionalisation methods have their drawbacks, mainly because this type of modification can damage the nanomaterial’s surface [7]. This surface damage can lead to a loss in the CNM’s unique electronic and physical properties, which may be essential to the nanocarrier’s effectiveness.

To circumnavigate issues associated with covalent modification, noncovalent functionalisation methods have been employed to attach components to the CNM. In this case, interactions, such as π–π stacking and hydrophobic interactions, are used to bind a molecule to the CNM surface [18].

A crucial aspect to consider when attaching drug molecules to a CNM is the drug release mechanism, which includes factors such as the release trigger and drug release profile, ensuring controlled and targeted delivery. Noncovalent attachment is particularly suited to reversible drug binding and can be utilised to design pH-responsive [19], redox-responsive [20], and NIR-responsive [21] drug delivery systems. Due to the acidity of tumour microenvironments, pH-sensitive systems are particularly relevant to cancer therapy [22]. This approach can be used to release the bound drug exclusively in the target tumour tissues, reducing unwanted side effects. Covalent strategies, such as drug attachment via hydrolysable ester bonds, have also been used for pH-responsive drug delivery, and often have the benefit of reduced drug leakage at neutral pH [23].

Herein, we present a database of nearly 200 CNM-based nanocarriers that have been utilised as drug delivery systems for clinically approved anticancer drugs. We curated this database through a comprehensive literature analysis, the details of which are provided in the following section. The entries are organised by drug type, and the composition, experimental results, drug loading and release metrics, and biological study models used are detailed. We also provide a critical analysis and discussion of the database and explore possible future research directions in the utilisation of CNM-based nanocarriers for anticancer drug delivery.

## 2. Methods and Metrics Used to Construct the Database

### 2.1. Preparation of the Database

This database is an in-depth overview of carbon nanomaterial (CNM)-based anticancer drug delivery systems. To construct the database, CAS SciFinder^n^ [24] was utilised as the data source. Combinations of keywords, such as “carbon nanomaterial”, “carbon nanotube”, “chemotherapy drug”, “anticancer drug”, and “doxorubicin”, were used to gather references, and the Boolean operators “AND” and “OR” were used to combine these search terms. Only English research articles that specifically focused on using CNMs to deliver clinically approved anticancer drugs were selected. The following information was extracted from each paper and entered into the database: (1) the anticancer drug used; (2) the composition of the nanocarrier system, including the CNM, and any targeting ligands, fluorophores, dispersants, etc., that were used; (3) the in vitro, in vivo, and ex vivo biological study models that the nanocarrier was tested on, including cell lines and animal breeds; (4) the drug loading and release metrics; these were taken only when explicitly given in the paper and were not calculated in this review; (5) the experimental results and observations, which were typically taken from the Conclusions section of each paper. The references were grouped based on the anticancer drug used, and the database was organised by sorting these drugs alphabetically.

### 2.2. Drug Loading and Release Metrics

The therapeutic efficacy of a nanocarrier system depends on its ability to absorb and release anticancer drugs; as such, quantitative metrics are needed to measure these systems. Such metrics are used to describe and compare the drug loading and release capabilities of different nanocarrier systems in the database.

The drug loading content (DLC) describes the amount of drug loaded onto the nanocarrier (Equation (1)). It is important to note that whilst most studies use the total mass of the nanocarrier (the CNM base, plus the drug, plus any other components), some studies just use the mass of the CNM itself [25], which leads to artificially higher DLC values.
(1)$$\mathrm{DLC}\;\left(\mathrm{wt}\%\right)=\frac{\mathrm{mass}\;\mathrm{of}\;\mathrm{drug}\;\mathrm{bound}\;\mathrm{to}\;\mathrm{nanocarrier}}{\mathrm{total}\;\mathrm{mass}\;\mathrm{of}\;\mathrm{nanocarrier}}\times 100$$

The drug loading efficiency (DLE), sometimes called the encapsulation or entrapment efficiency, is a measure of the effectiveness of the drug loading process and not a quantitative measure of the drug content (Equation (2)).
(2)$$\mathrm{DLE}\;\left(\mathrm{wt}\%\right)=\frac{\mathrm{mass}\;\mathrm{of}\;\mathrm{drug}\;\mathrm{bound}\;\mathrm{to}\;\mathrm{nanocarrier}}{\mathrm{total}\;\mathrm{mass}\;\mathrm{of}\;\mathrm{drug}\;\mathrm{added}}\times 100$$

The drug release efficiency (DRE) quantifies the cumulative release of a therapeutic agent from the nanocarrier (Equation (3)). This is the total amount of bound drug released throughout the experiment.
(3)$$\mathrm{DRE}\;\left(\mathrm{wt}\%\right)=\frac{\mathrm{total}\;\mathrm{mass}\;\mathrm{of}\;\mathrm{drug}\;\mathrm{released}}{\mathrm{mass}\;\mathrm{of}\;\mathrm{drug}\;\mathrm{bound}\;\mathrm{to}\;\mathrm{nanocarrier}}\times 100$$

## 3. Database of Carbon-Nanomaterial-Based Cancer Therapeutics

Herein, we present a database of CNM-based nanocarrier systems that transport clinically approved anticancer drugs, seen in Table 1 The database includes the composition of the nanocarrier, the in vitro and in vivo biological models the system was tested on, the drug loading and release metrics, and a summary of the experimental results. The database is organized alphabetically by the anticancer drug used in the formulation; an index can be seen in Table 2.

## 4. Discussion

A total of 38 approved anticancer drugs were used in CNM-based nanocarriers in the literature, a breakdown of which can be seen in Figure 1. The anticancer drugs were further classified according to their mechanism of action, and the prevalence of each class is shown in Figure 1. The classes are as follows: (1) Alkylating agents—these work by adding alkyl groups to DNA, which can lead to DNA strand breaks and inhibit DNA replication; (2) Antimetabolites—these interfere with the synthesis of DNA, RNA, and proteins by mimicking essential cellular metabolites; (3) Natural products—this category consists of chemotherapeutic agents derived from natural sources, such as plants, microorganisms, or marine organisms. These agents often target specific aspects of cell division or DNA replication; (4) Hormone therapies—these target hormone-dependent cancers by interfering with the action of specific hormones or hormone receptors; (5) Antimicrotubule agents—these drugs target the microtubules, which play an essential role in cell division. By disrupting the formation or function of microtubules, these agents can inhibit cell division and lead to cancer cell death; (6) Miscellaneous agents—these include chemotherapeutic drugs that do not fit neatly into any of the other categories. The most diverse class of anticancer drugs used were alkylating agents, which is not surprising, as this class includes many Pt-based drugs, which can be easily complexed with oxidized CNMs. The miscellaneous-agent section contained five tyrosine kinase inhibitors, indicating the popularity of this class of drug. Hormone therapies were the least popular class of chemotherapy agents, with only three entries. 

For each of the anticancer drugs in the database, several CNM-based nanocarriers have been investigated for their use in drug delivery. A total of 191 examples of CNM-based nanocarrier systems were found in the literature, many of which displayed higher anticancer efficacy with reduced side effects. As discussed in the introduction, the ease of functionalisation of CNM surfaces (graphene and CNT in particular) offers many different approaches to developing nanocarriers. A huge number of ligands were found to be used for drug delivery in the literature. These included polymers, such as PEG, which offer biocompatibility, water solubility, and reduced aggregation in situ [65,96], and biomolecules, such as folic acid, which enable the active targeting of folate receptors on tumour cells [177]. Other commonly used ligands were fluorescent agents, an example of which is Alexa Fluor, which is used for the fluorescent imaging of tumour cells [160], and peptides and proteins, offering improved bioavailability and stability [84]). Many of these approaches combined in the nanocarriers found in the literature show the complexity and breadth of options available to use.

The numerical analysis of the database shown in Figure 2 reveals that graphene (GO in particular) is the most popular class of CNMs to be incorporated into these systems, likely because it is one of the most well-established carbon nanomaterials. Over the years, a catalogue of functionalisation procedures has been developed, allowing for a range of moieties to be attached to the material’s surface [201]. The flat, aromatic surface of graphene lends itself excellently to π–π stacking, which allows for the easy noncovalent stacking of drug molecules. Graphene is also a PTT agent, a property that can be used to bolster the effectiveness of chemotherapy [14]. This is beneficial for the nanocarrier developed in [188], where free TAM is more efficacious than the nanocarrier-bound drug but the PTT potential of rGO makes it an attractive combinatorial therapy. Nonfunctionalised GO and rGO were found to successfully deliver anticancer drugs. GO and rGO showed pH- and NIR-triggered releases in some cases [32,132,148]. This offers a site-specific release of the drug, as the pH in tumour cells is typically lower than that in healthy cells, and NIR offers similar delivery.

CNTs came second in terms of popularity, which is surprising, as they are the oldest and most well-studied class of CNMs. This could be due to their tendency to aggregate into bundles in aqueous solutions, which could affect their biocompatibility. CNTs also do not have much in the way of intrinsic therapeutic or imaging properties; however, they do have large surface areas for drug loading. One example of pristine CNTs shows pH-dependent release and PTT [78]. Oxidised MWCNTs were found to be more toxic to healthy cells than cancer cells, which shows a need for further functionalisation [155]. The lack of control of intracellular accumulation also highlights the need for the attachment of targeting ligands to these systems.

Carbon dots, including GQDs, were the third most popular CNM to be used in these systems. The carbon allotropes that make up this new and rising class are tiny, even compared to other CNMs; this allows them to penetrate deep into cells to deliver drug molecules. They are even small enough to cross the blood–brain barrier [202], and their excellent biocompatibility [203] and ease of production in many cases [204] make them excellent nanocarrier scaffolds.

Many types of carbon dots are also water-soluble and have fluorescence imaging capabilities in the pristine form, for dual imaging and drug delivery [175]. Other types of CNMs, such as fullerenes, NDs, CNOs, HMCS, and CNHs, may not be as popular, but they show potential as nanocarriers due to their unique properties. For example, NDs offer improved biocompatibility over other CNMs [114,166], whilst fullerenes and CNOs display ease of functionalisation, narrow PDI, and ease of production [23,59].

In terms of drug moieties, antimicrotubule agents were the most popular payloads for these nanocarriers (Figure 2). This is due to the presence of multiple aromatic rings in these molecules, which facilitate noncovalent attachment to the CNM surface.

Anthracyclines, such as doxorubicin and epirubicin, are particularly popular. Alkylating agents are also quite popular, with platinum-based drugs often being complexed to the surface of the nanomaterial host. This is a particularly popular strategy with highly oxidised materials, such as GO, as the Pt can complex directly with oxygen-containing functional groups. Smaller hydrophilic organic molecules that lack any aromatic rings, such as those in the hormone therapy class, tend to not be so popular in CNM nanocarrier systems due to the lack of noncovalent interactions with the host CNM.

Designing systems that incorporate the intrinsic properties of CNMs allows for additional capabilities without having to chemically modify the material. The fluorescence of CDs and GQDs have proven to be useful for cellular imaging and tracking experiments [36,74,88]. Certain CNMs may also be utilised for killing cancer cells; for example, graphene has been used as a PTT agent, bolstering the effect of traditional chemotherapy [14,148]. Utilising this synergistic approach means that lower amounts of toxic chemotherapy drugs can be given to achieve the same therapeutic effect. The nπ* state of a CNM is essential for its intrinsic photothermal properties, and this state can be modulated by the addition of dopants (such as nitrogen) to the CNM [205]. Strong light absorption is required for a material to display photothermal properties, and a high photothermal conversion efficiency (η) is needed for a nanocarrier to be an effective PTT agent. For example, Forte et al. achieved photothermal-triggered drug release using a carbonised polymer dot-based system with an η value of 67.9% [206].

In general, systems that incorporate combination therapies exhibit some of the strongest anticancer effects due to the synergistic effects of PTT, PDT, single-drug and combination chemotherapy, or immunotherapy [67,129,169,190]. CNMs are the perfect class of nanoparticles for this approach, as they are easy to modify both covalently and noncovalently, with a range of functionalisation approaches available, allowing for the attachment of many different therapeutic agents. This, combined with the intrinsic properties of CNMs, can be leveraged to construct a range of nanocarrier systems.

The DLCs, DLEs, and DREs of nanocarriers are given as the number of entries above and below 50% in Figure 3. This was performed to make the entries more comparable given the differences in the sample sizes. A total of 152 out of 191 nanocarriers found in the literature had DLC, DLE, and/ or DRE data, and where DLC data were given, less than 35% of the nanocarriers were found to display values above 50%. This is similar for all CNMs, and surprisingly, graphene is the lowest, with only 22% of nanocarriers above 50% DLC. A much greater proportion of nanocarriers show DLEs above 50%, with fullerenes showing the lowest percentage of DLE values above 50%, and nanohorns displaying the highest at 100%; however, only one entry was available. This indicates that drug loading is an efficient process. In general, either a DLC or DLE value was given, with the DLE higher than the DLC.

The DRE, where given, was found to be quite high, on average, for all CNMs, with NDs displaying the lowest values; however, the sample size for this material was small compared to those for graphene and CNTs. Graphene-based nanocarriers incorporating Fe_3_O_4_ showed particularly high loading and release: [47,85,193]. GO-COOH also displayed high loading and release properties [133]. A lot of CNT-based systems with very high loading and release were observed; for example, the FA-PEG-bis-amine MWCNT system displayed 99% DLE and 90% DRE [27], SWCNTs showed 94% DLE and 93% DRE [78], and CNT-PEG displayed 95% DLC and 100% DRE [186].

For carbon dots, the CS-coated Fe3O4–NH2/GQD hybrid displayed 90% DLC + 84% DRE [33], whereas the GQD-HA system showed 98% DLE + 100% DRE [83], and the FA-CD-GOx nanocarrier yielded 82% DLE + 95% DRE [13]. Whilst there were much fewer fullerene systems with high drug loading/release metrics, acylated C60 fullerene displayed 81% DLC and 84% DRE [99]. A single ND system yielded 87% DLE and 80% DRE [166], whereas NHs and NPs do not show high combined loading and release.

The biocompatibility of these formulations must be further investigated, as certain CNMs are known to be toxic [207]. On the one hand, in the case of CNTs, the pristine form is toxic in mice and is dependent on the types of CNTs present [208]. On the other hand, pristine fullerenes such as C60 show no apparent toxicity, whereas some functionalised derivatives are highly toxic [209]. However, as previously discussed, the breadth of functionalisation methods and biocompatible ligands available to modify the surface chemistries of CNMs offers a variety of routes for overcoming this issue. The leakage of a drug at physiological pH is another issue that must be addressed in many systems, as toxic side effects are induced in vivo when the drug is released at neutral pH.

## 5. Conclusions and Future Directions

Overall, CNMs are incredibly versatile materials that can be used as both the foundation of nanocarrier systems and as therapeutic agents themselves. These systems can be designed to detect, image, and treat a range of tumours, from colorectal, brain, breast, liver, and stomach cancers. CNTs and graphene (GO in particular) were by far the most popular CNMs used in these systems due to their small size, high surface area, and ease of functionalisation. Other CNMs, such as carbon dots, are also growing in popularity due to their unique properties. A huge range of molecules, such as targeting ligands, fluorophores, dispersants, and drugs, can be easily attached to CNM surfaces, allowing for the construction of complex nanosystems.

Extensive in vivo biological work needs to be undertaken to fully understand the toxicity of these systems towards animals, and to overcome the regulatory hurdles needed to move these treatments into clinical trials.

The trend of designing theragnostic systems that incorporate the intrinsic properties of CNMs (such as PTT and fluorescence imaging) will likely be seen more in the future, as it allows for additional capabilities without damaging the CNM itself. Nanocarriers that leverage combinations of different therapies displayed the most potent anticancer effects, and therefore these systems will likely grow in popularity in the coming years.

## Figures and Tables

**Figure 1 pharmaceutics-15-01545-f001:**
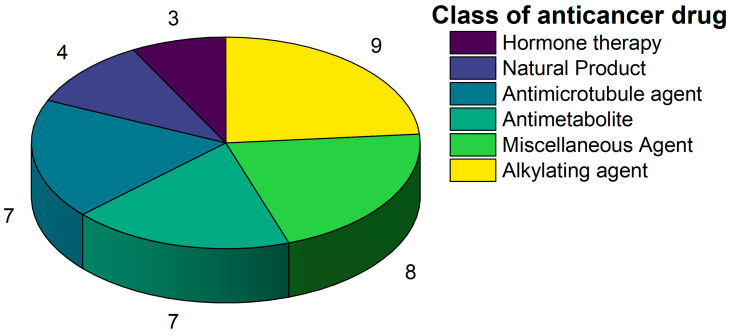
Number of anticancer drugs according to class.

**Figure 2 pharmaceutics-15-01545-f002:**
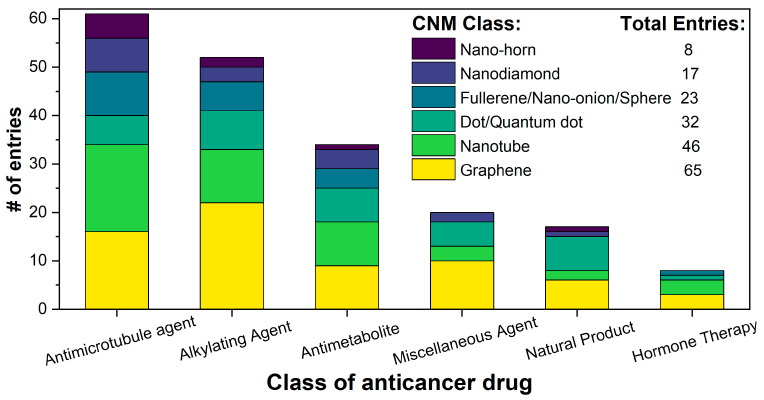
A stacked bar chart depicting the number of database entries for each CNM type under each class of anticancer drug.

**Figure 3 pharmaceutics-15-01545-f003:**
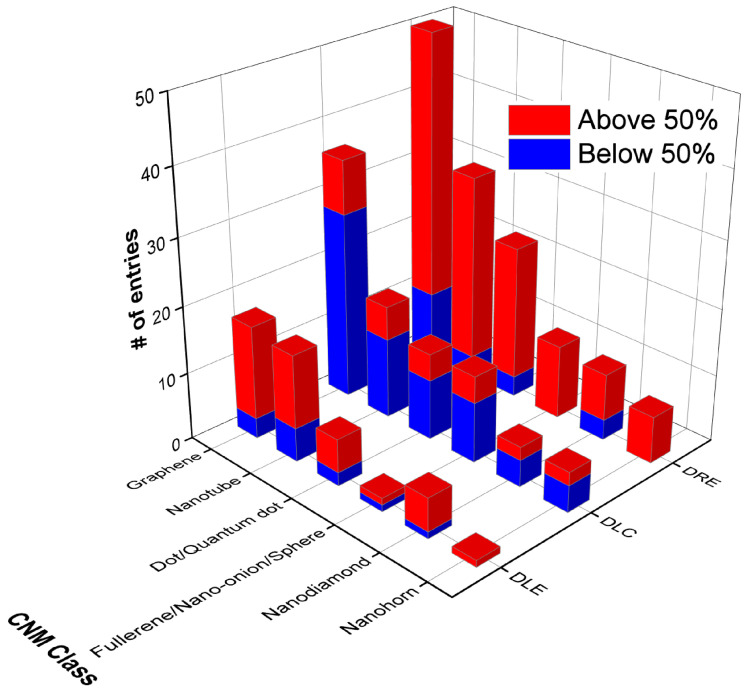
The number of entries that have DLEs, DLCs, or DREs above or below 50%, grouped by CNM class.

**Table 1 pharmaceutics-15-01545-t001:** Database of CNM-based anticancer nanocarriers.

Chemotherapeutic—Drug Class	CNM-Based Nanocarrier	Biological Study Models	Drug Loading and Release Metrics	Experimental Results	Ref.
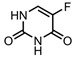 5-fluorouracil (5-FU)—pyrimidine antimetabolite	CS-carbon quantum dot (CQD)-Apt	in vitro: MCF-7 cells	32% DLC, approximately 100% DRE, and pH-sensitive, controlled 5-FU release	The unloaded nanocarrier is biocompatible, and the use of an aptamer increases uptake and cytotoxicity in breast cancer cells.	[26]
FA-PEG-bis-amine multiwalled carbon nanotube (MWCNT)	in vitro: MCF-7 cells	99% DLE, ~90% DRE, with pH-triggered drug release sustained over 900 min	This nanocarrier increases circulation time, half-life, and accumulation of 5-FU in target tissues, and this leads to the effective killing of breast cancer cells in vitro.	[27]
	CS/Au/MWCNT	in vitro: MCF-7 cells	43% DLC, 59% DRE, with prolonged, sustained drug release	Reduced potential side effects and increased efficacy compared to free 5-FU were observed. A reduction in cancer cell viability was observed at low nanocarrier concentrations.	[28]
	Nanodiamond (ND)-ADH	in vitro: MCF-7 and HepG2 cells	88% DLE, 35% DRE, with pH-mediated, sustained drug release	This nanocarrier showed potent anticancer effects with low haemolytic toxicity in human blood.	[29]
	Mesoporous carbon nanoframe (mCNF)	in vitro: HeLa cells	31% DLC, 80% DRE, with dual pH/NIR-triggered drug release	This system displayed excellent photothermal efficiency with the NIR pulse-triggered burst release of 5-FU. The photothermal conversion efficiency of this system was found to be 21%. This synergistic chemo–photothermal therapy combined with photoacoustic imaging capabilities can effectively treat cancer in vitro.	[30]
	PEG-C_60_ fullerene–alanine	in vitro: MCF-7 and BGC-823 cells	1% DLC, with no quantitative drug release data	The unloaded nanocarrier displays good biocompatibility and the system is stable in murine serum for over 24 h. This formulation results in the significantly better inhibition of cancer cells compared to free 5-FU.	[31]
	Graphene oxide (GO)	in vitro: A549 cells	31% DLC, 35% DRE, with pH-triggered drug release	The blank nanocarrier is biocompatible and the loaded system improved the stability of 5-FU.	[32]
	CS-coated Fe_3_O_4_–NH_2_/graphene quantum dot (GQD) nanohybrid	in vitro: A549 cells	90% DLC, 84% DRE, with pH-dependent drug release	This system has magnetic resonance/fluorescence imaging capabilities and displayed significantly higher cytotoxicity than free 5-FU, whilst the unloaded nanocarrier is biocompatible.	[33]
	HPMC/GO	in vitro: Vero, HepG2, and A549 cells		No quantitative drug loading/release studies were performed. The blank nanocarrier displays high biocompatibility in normal cells, whilst the drug-loaded system displays a higher antitumour efficacy than free 5-FU. A green synthesis method was used.	[34]
	TAU-GO	in vitro: HepG2 cells; in vivo: SD rats	50% DLE, 90% DRE, with pH-triggered 5-FU release	This biocompatible nanocarrier improved the circulation time and anticancer efficacy of 5-FU.	[35]
	Carbon dot (CD)-BT	in vitro: MCF-7, HeLa, and HEK-296 cells	35% DLE, 81% DRE, with pH-triggered drug release	An initial burst of 5-FU is followed by sustained release; this nanocarrier also displays fluorescence imaging capabilities. BT-mediated targeting of cancer cells resulted in high cytotoxicity towards neoplastic cells and increased cellular uptake due to biotin-receptor-mediated endocytosis.	[36]
	N-doped mesoporous carbon sphere (NMCS)-DSPE-PEG	in vitro: B16F0 cells	38% DLC, 78% DRE, with dual pH/NIR-triggered drug release	This nanocarrier produces reactive oxygen species when irradiated with an NIR laser, and the resulting PDT/PTT/chemotherapeutic combination therapy effectively kills melanoma cells much more efficiently than 5-FU alone.	[37]
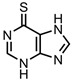 6-mercaptopurine (6-MP)—purine antimetabolite	CD-BT	in vitro: CHO, MCF-7, and HepG2 cells	5% DLC, 79% DRE, with dual pH- and redox-sensitive drug release	Comparable anticancer activity to free 6-MP (in cancer cells) with much lower cytotoxicity (in healthy cells). A GSH-sensitive carbonyl vinyl sulphide group was used to bind 6-MP to BT.	[38]
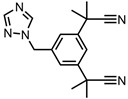 Anastrozole—aromataseinhibitor	GO-Fe nanoparticles	in vitro: MCF-7 cells	84% DLE	No qualitative drug release data shown. The system displays higher cytotoxicity than the free drug for cancer cells, and it has magnetic properties.	[39]
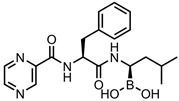 Bortezomib—proteasomeinhibitor	CD-CuS NPs-MMT7	in vitro: U251 MG cells		Synergistic drug delivery and PTT platform that specifically targets cancer cells. No qualitative drug loading/release data shown. This innovative nanocarrier combines immune system evasion capabilities with the enhanced suppression of tumour growth and metastasis to provide excellent control over cancer growth and metastasis.	[12]
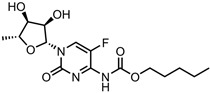 Capecitabine—pyrimidineantimetabolite	Single-walled carbon nanotube (SWCNT)-FL-FA-NCC	in vitro: Caco-2/TC7 cells		No quantitative drug loading/release data shown. This nanocarrier is nontoxic and has fluorescence imaging capabilities. The effective targeting of colon cancer cells leads to an increase in anticancer activity compared to the free drug.	[40]
oxiSWCNT-CS-FA	in vitro: COLO320DM and HT29 cells; in vivo: albino rabbits	94% DLE, 89% DRE	An increase in cytotoxicity compared to free drug was noticed during in vitro experiments. The capsule formulation of this nanocarrier is exclusively released in the colon in vivo, avoiding premature release in the stomach. Active targeting of cancer cells was achieved via the FA-targeting ligand.	[41]
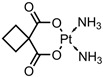 Carboplatin (CP)—DNAalkylating agent	GO-PAMAM	in vitro: hMSC and HeLa cells		No quantitative drug loading/release data shown.The 100 nm width GO (unloaded) was found to be the least toxic. This system displayed enhanced anticancer activity compared to free CP, with decreased cytotoxicity.	[42]
GO-gelatine	in vitro: IMR-32 and hMSC cells	99% DLE, with no quantitative drug release data shown	This formulation displays effective CP delivery and uptake in vitro, resulting in a higher potency than free CP. Excellent biocompatibility and stability were observed in vitro.	[43]
	oxiMWCNT-HA	in vitro: TC–1 and NIH/3T3 cells	No quantitative drug loading/release data shown	This system displayed the selective uptake and targeting of cancer cells over healthy cells, resulting in significantly higher cytotoxic effects in neoplastic cells and lower side effects in healthy cells.	[44]
	Aminated MWCNT	in vitro: MDA-MB-23 and MCF-12A	89% DLC, 21% DRE	This formulation provided increased cancer cell death compared to free CP and killed cells via an ROS-triggered autophagy mechanism.	[45]
	FA-CDT-C_60_ fullerene	in vitro: HeLa, HeLa-RFP, and A549 cells; in vivo: Danio rerio, both healthy and bearing HeLa tumours	37% DLC, ~80% DRE, with pH-triggered drug release	This system displayed increased anticancer effects compared to the free drug alone due to the active targeting of folate-receptor-overexpressing cancer cells and improved cellular uptake. Low toxicity and improved antitumour effects compared to the free drug were also seen in vivo.	[46]
	CS-Fe_3_O_4_-GO	in vitro: HepG2 and MCF-7 cells	74% DLE, 90% DRE, with pH-triggered drug release	A very high amount of CP was released at neutral pH.Despite this, an increase in CP potency and a reduction in systemic toxicity was observed.	[47]
	GO-CS-FA	in vitro: LX-2 and SKOV3 cells	14% DLC, ~90% DRE	CP release was similar in neutral and acidic environments; hence, this system is unsuitable for pH-triggered drug release via noncovalent drug attachment. The system showed slightly lower cancer cell inhibition than free CP.	[48]
	GO-Fe_3_O_4_-PANI	in vitro: SMMC-7721, HepG2, and HL-7702 cells	~95% DRE; the qualitative drug loading data provided does not account for unbound CP that was removed	The blank nanocarrier showed efficient cellular uptake and negligible cytotoxicity. This nanocarrier has magnetic properties and pH-triggered drug release.	[49]
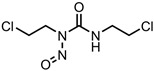 Carmustine—DNAalkylating agent	GO-PAA	in vitro: GL261 cells	19% DLC, with no quantitative drug release data shown	A significant increase in half-life, >70% decrease in IC_50_ value, and 30% increase in inter-strand DNA crosslinking was observed compared to the free drug in vitro.	[50]
N-doped carbon nanotube (CNT) sponges	in vitro: rat astrocytes, C6, RG2, and U87 cells	~90% DRE, with no quantitative drug loading data	This nanocarrier displayed similar cytotoxicity to the free drug, with a sustained-release profile. The sponges appear to be more biocompatible than CNTs alone; hence, the blank nanocarrier showed low cytotoxicity, whilst the drug-loaded system displayed strong anticancer effects.	[51]
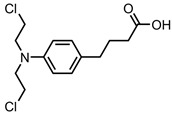 Chlorambucil—DNAalkylating agent	Reduced graphene oxide (rGO)-FA-gelatine	in vitro: Siha cells	35% DLC, 82% DRE, with pH-triggered drug release	A significant decrease in IC_50_ value compared to the free drug was observed. The use of gelatine facilitated sustained drug release. This system is a promising treatment for cervical adenocarcinoma.	[52]
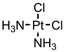 Cisplatin (CisP)—DNAalkylating agent	GO-Ala	in vitro: MCF-7 and HepG2 cells	4% DLC and ~70% DRE, with sustained drug release at neutral pH	The blank nanocarrier is biocompatible, whilst the CisP-loaded material is effective at killing cancer cells in vitro.	[53]
Oxidised carbon nanohorn (CNH)	in vitro: NCI-H460	Approximately 1% DLC and 80% DRE, with sustained CisP release	This nanosystem displayed similar anticancer effects to the free drug in vitro.	[54]
	CNH-quantum dot	in vitro: AY-27 cells	18% DLC and 70% DRE, with a sustained drug release profile	This theragnostic system displayed a significant reduction in anticancer potency compared to free CisP; however, it has imaging capabilities arising from the inclusion of CdSe quantum dots.	[55]
	Ultra-short SWCNT	in vivo: SCID/beige mice bearing MCF-7, BCM-4272, and MDA-MB-231 tumours		This nanoformulation effectively treated CisP-resistant breast cancer in a xenograft mouse model. The nanocarrier also displays an enhanced circulation time and increased tumour localisation, leading to increased potency compared to the free drug.	[56]
Silane-modified ND	in vitro: HeLa cells		This unique system has a Pt loading of 0.25 mmol/g ND, and CisP is not released from the conjugate. Despite this, the system displayed a similar IC_50_ value to CisP. The main advantage of this system is the prevention of CisP isomerisation, leading to enhanced aqueous stability. No quantitative drug loading/release data were given.	[15]
	SA/ND	in vitro: HepG2, HeLa, A549, and RAW264.7 cells		This sustained-release drug platform improved the CisP accumulation in cancer cells, with improved drug safety. Whilst no quantitative drug loading or release data were given, no change in the antitumour mechanism was observed compared to the free drug.	[57]
	EGF-ND	in vitro: HepG2 cells	1% DLC, with no quantitative drug release data shown	This nanoformulation was capable of selectively killing liver cancer cells and displayed increased potency compared to free CisP. This was due to the EGF-mediated targeting of cancer cells. In addition, the NDs are probes for 3D Raman microscopy imaging. This nanocarrier system induces morphological changes in cancer cells, resulting in higher surface areas for CisP absorption with a lower risk of adverse side effects.	[58]
	C_60_ fullerene	in vivo: BALB/c mice	50% DLC, with no quantitative drug release data shown	A two-fold decrease in systemic toxicity (LD_50_) compared to free CisP was observed. Specifically, the nanocarrier decreased drug-induced leukopenia, anaemia, thrombocytosis, and inflammation.	[59]
	C_60_ fullerene	in vitro: LLC cells	No quantitative drug loading or release data given	A 4.5× decrease in IC_50_ value compared to the free drug was observed in vivo. The fullerene itself was found to increase the cellular uptake and accumulation of CisP.	[60]
	oxiC_60_ fullerene	in vitro: L929 cells	16% DLC, 60% DRE, with pH-triggered drug release	This nontoxic nanocarrier displayed outstanding fluorescence properties for cellular imaging experiments.	[61]
	C_60_ fullerene	in vitro: HCT-116, HeLa, HL-60, HL-60/adr, and HL-60/vinc cells; in vivo: C57BL/6J mice bearing LLC tumours	No quantitative drug loading/release data shown	This nanocarrier killed chemotherapy-resistant leukaemia cells in vitro and exhibited effective lung cancer tumour growth inhibition in vivo. Molecular docking studies suggested that the fullerene binds to proteins involved in chemotherapy resistance.	[62]
	CS-GO	in vitro: HeLa cells	71% DLE, 88% DRE, with pH-triggered drug release	The functionalisation of GO with CS and CisP dramatically reduced protein binding. This biocompatible nanocarrier triggered apoptosis in drug-resistant cancer cells.	[63]
CQD-GE11-DOX	in vitro: CNE-2 cells; in vivo: BALB/c mice bearing CNE-2 tumours	5% DLC and 57% DRE, with pH-triggered drug release	Specific tumour targeting and inhibition was observed in vivo, and the effective killing of nasopharyngeal carcinoma cells was exhibited in vitro. This nanocarrier has fluorescence imaging capabilities and showed no obvious side effects in vivo.	[64]
	GO-PEG	in vitro: MG63, SAOS-2, U2-OS, MDA-MB-231, MDA-MB468, U118, and U87 cells	64% DLE, with no quantitative drug release data shown and redox-sensitive drug delivery using a CisP prodrug	This nanoformulation displayed high uptake and proliferation inhibition in osteosarcoma cells, and effective internalisation, but reduced potency in glioblastoma cells. This system is capable of inhibiting cell migration in highly invasive breast carcinoma.	[65]
	MnO_2_-GO-Ce6-HA	in vitro: MDA-MB-231 and RLE-6TN cells; in vivo: BALB/c mice bearing MDA-MB-231 tumours	7% DLC and 60% DRE, with pH-triggered drug release	Combination therapy of (1) MnO_2_ to regulate the tumour microenvironment, enhancing the anticancer effect of (2) CisP chemotherapy and (3) Ce6 PDT. The incorporation of HA facilitates tumour targeting for a true theragnostic system. This system also shows excellent biocompatibility and antitumour efficacy in vivo.	[66]
	rGO-PHEMA-DOX	in vitro: MCF-7 cells	82% DLE and 64% DRE, with pH-triggered drug release	A significant decrease in IC50 value compared to free CisP and DOX was observed. This biocompatible nanocarrier displayed efficient cellular uptake and a synergistic effect between the two loaded drugs, resulting in the effective killing of breast cancer cells.	[67]
	Fe_3_O_4_-rGO-PHEMA-MET	in vitro: HepG2 and Caco-2 cells; in vivo: BALB/c mice, both healthy and bearing HepG2 tumours	82% DLE, 60% DRE, with pH-triggered drug release	No side effects and potent antitumour efficacy was noted in vivo. This highly biocompatible nanosystem effectively killed hepatocellular carcinoma in vitro.	[68]
	GO-PEG-DOX	in vitro: CAL-27, L929, and MCF-7 cells; in vivo: nude mice carrying CAL-27 tumours	37% DLC and 65% DRE, with pH-triggered drug release	A 2× increase in cancer cell apoptosis and necrosis compared to the single-drug-loaded nanocarrier was observed. An attenuation of toxicity and enhanced anticancer effects compared to free DOX/CisP were observed.	[69]
	S-doped CD	in vitro: A2780 and A2780 cells	No quantitative drug loading/release data shown	The unloaded CDs were found to be biocompatible and could interact with proteins and lipids on the surfaces of cancer cells. A similar IC_50_ value to the free drug was seen in normal ovarian cancer cells, and the nanoformulation could kill drug-resistant cancer cells.	[70]
	CD-iRGD	in vitro: A549, HUVEC, and HEL-299	No quantitative drug loading/release data shown	This nanocarrier destroyed lung cancer cells whilst leaving healthy cells unharmed.	[71]
	MWCNT	in vitro: A549 and A549/DDP cells; in vivo: BALB/c mice carrying A549/DDP tumours	No quantitative drug loading/release data given	The unloaded MWCNTs were found to be biocompatible, whilst the loaded nanocarrier had higher cytotoxicity against cancer cells than free CisP. This nanoformulation could effectively treat a drug-resistant lung cancer in vivo model.	[72]
	PDA CD-anti-EpPCAN	in vitro: HepG2 cells; in vivo: BALB/c mice bearing HepG2 tumours		This synergistic nanocarrier combined a cisplatin prodrug with significant PTT and fluorescence imaging capabilities for effective image-guided chemo–photothermal therapy. This biocompatible system exhibited excellent antitumour effects in vitro and in vivo.	[73]
	CD-PEG	in vitro: GES-1 and MGC-803 cells	5% DLC, with no quantitative drug release data given and pH/redox-mediated drug release achieved via a hydrolysable benzoic imine bond	A CisP prodrug was used, and the resulting system had comparable anticancer efficacy to free CisP, with reduced side effects. This system also exhibited fluorescence imaging capabilities.	[74]
	CD-PEG	in vitro: A549, HUVEC, and HEL-299 cells	No quantitative drug loading data given, 86% DRE and redox-sensitive drug release	This fluorescent nanocarrier could effectively kill cancer cells whilst leaving healthy cells unharmed.	[75]
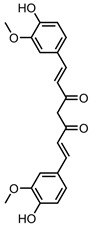 Curcumin (CUR)—ROS scavenger and lipid peroxidation inhibitor	Fluorinated GO-LIN	in vitro: MCF-7 and MCF10A cells; in vivo: BALB/c mice with 4T1-induced breast cancer	61% DLC, 95% DRE, with pH-sensitive drug release	This simple, low-cost system acts as a contrast agent for magnetic resonance imaging. It also displays improved cytotoxicity, tumour suppression, and reduced side effects compared to free CUR due to effective cancer cell targeting.	[76]
GQD-GlcN	in vitro: MCF-7 cells	No quantitative drug loading information, 37% DRE with pH-triggered, sustained drug release	The unloaded nanocarrier is biocompatible, whilst the loaded system exhibits effective cancer cell targeting and internalisation in vitro. This system also has fluorescence imaging capabilities.	[77]
SWCNT	in vitro: PC-3 cells	94% DLE, 95% DRE, with pH-triggered drug release	The combination of efficient CUR delivery and SWCNT-mediated PTT successfully inhibited tumour cell growth. This nanocarrier also reduced CUR biodegradation and increased its solubility.	[78]
SWCNT-PC-PVP	in vitro: PC-3 and S180 cells; in vivo: Kunming mice bearing S180 tumours		No quantitative drug loading/release data. This biocompatible nanocarrier increased CUR cellular uptake, plasma concentration, and bioavailability. The system overcomes the main barrier to the low anticancer effect of free CUR (low plasma concentration) whilst displaying low in vivo toxicity. This is a combination therapy with the SWCNT-mediated photothermal ablation of cancer cells.	[79]
	oxiND-ADH	in vitro: MCF-7 andHepG2 cells	93% DLE, 36% DRE, with pH-triggered, sustained drug release	The use of a pH-sensitive amide bond to bind CUR slows release and increases stability, resulting in potent cytotoxicity.	[29]
	Graphene oxide quantum dot (GOQD)-CS-PEG-MUC-1 aptamer	in vitro: MCF-7 and HT-2 cells	99% DLC, 64% DRE, with pH-responsive drug release	This system effectively targets MUC-1-overexpressing cancer cells whilst displaying photoluminescence imaging and cancer detection abilities. An increase in therapeutic efficacy and cellular uptake compared to free CUR and low haemolysis with human blood was observed with this system.	[19]
	CD-PNM	in vitro: SH-SY5Y cells	No qualitative CUR loading information, 82% DRE	This formulation resulted in a 10× enhancement of CUR solubility whilst displaying excellent photophysical properties and low toxicity.	[80]
	CD	in vitro: HepG2 and A549 cells	3% DLC, ~90% DRE, with pH-mediated drug release	This cost-effective, photoluminescent nanocarrier is nontoxic to normal cells and displayed potent anticancer effects with enhanced CUR bioavailability and a small size.	[81]
	CoFe_2_O_4_/GO-ADH-CMC	in vitro: MDA-MB-231 and MCF-10A cells	2% DLC, 86% DRE, with pH-triggered, controlled drug release	A decrease in cancer cell viability compared to free CUR was noted when using this system.	[82]
	GQD-HA	in vitro: HeLa and L929 cells	98% DLE, ~100% DRE, with pH-triggered drug release	This nanoformulation displayed excellent anticancer activity compared to CUR alone and has no toxic effect on healthy cells. The system is also highly fluorescent when CUR is released.	[83]
	GO-BSA-AS1411 aptamer	in vitro: MCF-7 and SKBR3 cells	9% DLC, 70% DRE, with pH-triggered drug release	Efficient targeting of nucleolin-overexpressing MCF7 cancer cells was achieved, facilitated by aptamer attachment. This resulted in increased CUR antitumour activity. BSA decoration was found to improve nanocarrier biostability and slow CUR degradation.	[84]
	CS-Fe_3_O_4_-RGO	in vitro: MCF-7 cells	95% DLE, 96% DRE	This system successfully targeted and induced apoptosis in breast cancer cells. The superparamagnetic nanocarrier increased the rate of CUR delivery compared to the free drug.	[85]
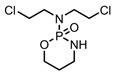 Cyclophosphamide—DNA alkylating agent	Hollow mesoporous carbon sphere (HMCS)	in vitro: CNE cells; in vivo: nude mice with CNE tumours	20% DLC, with no release experiments carried out with PTX	The HMCSs themselves exhibited strong antitumour effects through PTT.	[86]
PAA/PEG/CNT/MTX		Approximately 80% DRE, with no quantitative drug loading data shown	The system displayed dual pH- and temperature-triggered drug release, with an initial burst of drug followed by sustained delivery.	[87]
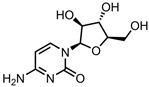 Cytarabine—DNA polymerase inhibitor	GQD-CS		73% DLE, 72% DRE, with pH-sensitive drug release via a hydrolysable amide linkage	This nanosystem possesses remarkable fluorescence stability, and CS wrapping enhanced the water solubility of this system.	[88]
Au/GQD/MPA/PEI	in vitro: HL-60 cells	68% DLE, 78% DRE, with dual pH- and NIR-triggered drug release	Cytarabine was attached via charge–dipole interactions. The chemo–photothermal combination therapy had higher efficacy than PTT alone.	[21]
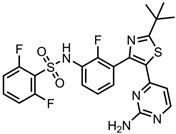 Dabrafenib—reversible ATP-competitive kinase inhibitor	GO-BSA	in vitro: A375, HDF, SKmel28, SKmel23, MelJuSo, MNT-1, and NHEM cells	No quantitative drug loading/release data, with pH-triggered drug release	The potency of dabrafenib was retained, with effective BRAF and HDAC inhibition in human melanoma cells.	[89]
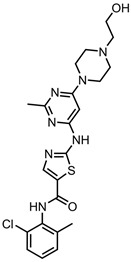 Dasatinib—tyrosine kinaseinhibitor	PLA-PGA-PEG-CNT	in vitro: U-87 cells	4% DLC, ~65% DRE	This nanocarrier system was synthesised via a simple one-pot method and demonstrated improved therapeutic efficacy compared to the free drug in vitro. The drug release profile of this system can be controlled by varying the composition of the polymer coating.	[90]
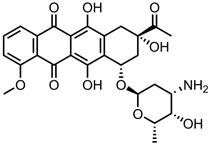 Daunorubicin (DNR)—topoisomerase II inhibitor	P-gp-GGN	in vitro: adriamycin-resistant leukaemia cell lines KA and K562/A02 cells; in vivo: KA nude mice with drug-resistant leukaemia-cell-induced tumours	32% DLC, roughly 45% DRE, with redox-triggered DNR release facilitated by increasing glutathione concentration	This nanocarrier overrides the cell‘s drug resistance to facilitate DNR uptake, resulting in a remarkable inhibition of tumour growth in vivo.	[20]
PLA/MWCNT/FE_3_O_4_	in vitro: K562 cells	96% DLE, roughly 55% DRE, with dual magnetic field- and pH-mediated drug release	The most effective killing of leukaemia cells was observed at a 20 µg/mL nanocarrier concentration.	[91]
	ND	in vitro: K562 cells	~95% DLC, most of the bound DNR was released	This formulation achieved a three-fold reduction in the IC-50 value compared to free DNR for the treatment of drug-resistant K562 cells.	[92]
	f-CNTs		94% DLE at 3:1 DNR:f-CNT ratio, with no quantitative drug release data provided	Hydroxylated CNTs provided the best DNR binding (through electrostatic interactions).	[93]
	Bi_2_MoO_6_/NH2-GO/PEG	in vitro: HUVEC and MCF-7 cells	33% DLC, 86% DRE, with pH-triggered drug release	DNR was selectively released in cancer cells. Haemolysis and coagulation tests prove system has no negative effects on the blood.	[94]
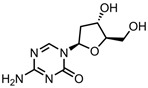 Decitabine—DNA methyltransferase inhibitor	A1-GO	in vitro: A549, NCI-H157, NCI-H520, NCI-H1299, NCI-H446, MCF-7, and HeLa cells	64% DLE, 75% DRE, with pH-dependent drug release	Specific recognition and targeting of lung cancer cells over other cancer cells was achieved by using the A1 aptamer. This system achieved much higher anticancer efficacy than the free drug.	[95]
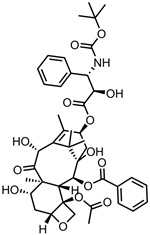 Docetaxel (DTX)—microtubule growth inhibitor	GO-PEG	in vitro: DU-145 cells	No quantitative drug loading or release data provided	This system was highly effective at killing prostate cancer cells due to a decrease in IC_50_ compared to free DTX. The nanocarrier displayed low dispersion stability in biological fluids.	[96]
RGD-CS-SWCNT	in vitro: A549 and MCF-7 cells; in vivo: BALB/c mice inoculated with A549 tumours	32% DLC, 68% DRE, with pH-triggered drug release	Significant drug uptake and growth inhibition in A549 cells was observed. The system entered cells via clathrin- and caveolin-mediated endocytosis and displayed strong tumour targeting, growth inhibition, and biosafety in vivo.	[97]
Oxi-carbon nano-horn (CNH)-PEG-mAb	in vitro: MCF-7; in vivo: ICR mice xenografted with H22 tumours	74% DLE, 59% DRE	The adsorption of DTX to the nanohorns was achieved via π–π stacking. Prolonged diffusion-controlled DTX release was achieved. The use of mAb resulted in the selective killing of cancer cells in vitro and in vivo and a lower IC50 and no significant side effects compared to free DTX in vivo. This nanocarrier also leveraged the enhanced permeability and retention effect.	[98]
	Acylated C_60_ fullerene	in vitro: MCF-7 and MDA-MB-23 cells; in vivo: Wistar rats	81% DLC, 84% DRE	This system achieved 4.2× higher bioavailability and 50% lower drug clearance compared to free DTX. This resulted in enhanced cancer cell cytotoxicity, low haemolysis, and high erythrocyte compatibility.	[99]
	Hexagonal nanostructured GO	in vitro: A549 cells	41% DLE and approximately 20% DRE, with pH-mediated drug release	The nanostructured material improved the drug loading capacity compared to pristine GO and displayed good biocompatibility.	[32]
	Carbon nanoparticle (CNP)-HIF-PLGA	in vitro: Walker256 cells; in vivo: rats xenografted with Walker256 tumours	16% DLC, with no qualitative release data shown and NIR-activated drug release	This nanocarrier displayed photothermal properties. The synergistic effect of chemotherapy and image-guided NIR PTT gave this system the ability to effectively target and treat metastatic lymph nodes both in vitro and in vivo, with minimal side effects.	[100]
	CNP-PLGA	in vitro: MDA-MB-231 and HUVEC cells; in vivo: New Zealand white rabbits bearing VX_2_ liver tumours	NIR-triggered drug release, with no quantitative drug loading/release data shown	This system relied on a combination of PTT and photoacoustic imaging to treat cancer in vitro and in vivo. Highly targeted drug delivery was achieved by transport through the lymphatic system to produce an excellent therapeutic effect on metastatic lymph nodes with favourable biocompatibility and biosafety.	[101]
	CNS	in vitro: MDA-MB 231 cells	92% DLE, with no quantitative release data shown	CS nanopores substantially assisted in drug loading to give this system favourable anticancer properties.	[102]
	Hydroxylated CNT-APA	in vitro: MDA MB-231 cells; in vivo: Wistar rats	51% DLE, with no quantitative drug release data shown and pH-triggered drug release in cancer cytosol	This formulation achieved a 2.8× enhancement in cytotoxicity and superior pharmacokinetics compared to free DTX, with substantial hemocompatibility with human blood and reduced side effects compared to the drug alone in vivo.	[103]
	RBC@GQD	in vitro: A549 cells; in vivo: CAnN.CgFoxn mice carrying A549-induced tumours; C57BL/6J mice with ALTS1C1 intracranial tumours	Approximately 40% DRE, with no qualitative drug loading data and NIR-triggered drug release	This system achieved an eight-fold increase in the accumulation in tumour tissues compared to the free drug. The synergy between the chemotherapy and photolytic properties of the nanocarrier allowed for deep penetration into tumours and effective treatment in vivo.	[104]
	C_60_ fullerene-APA	in vitro: MDA MB-231 cells; in vivo: Wistar rats	48% DLE, 96% DRE, with pH-triggered drug release	A substantial decrease in haemolysis (human blood) and protein binding (BSA) compared to free DTX was observed. The nanoformulation also increased bioavailability and potency compared to free DTX. Fullerenes display partial P-gp efflux inhibition.	[105]
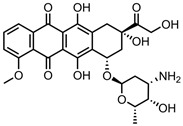 Doxorubicin (DOX)—topoisomerase II inhibitor	GO/PEI.Ac-FITC-PEG-LA	in vitro: SMMC-7721 and PIEC cells	85% DLC, ~80% DRE, with pH-dependent drug release	The nanocarrier shows specificity for ASGPR-receptor-containing cancer cells whilst retaining DOX therapeutic efficacy.	[106]
GO-PRM/SA	in vitro: MCF-7 cells	29% DLC, 49% DRE, with pH-dependent drug release	Protein adsorption in physiological environments was suppressed and the system showed enhanced cytotoxicity compared to GO-DOX alone.	[107]
	Tf/FA/GO/PF68	in vitro: SMMC-7721 and L-02 cells	96% DLC, 55% DRE, with pH-dependent drug release	The nanocarrier displayed low toxicity and high specificity due to the synergistic effect of Tf- and FA-targeting ligands on cancer cell targeting. The DOX-loaded nanosystem was not tested on healthy L-02 cells.	[108]
	GNRs/GO@PDA nanosheets	in vitro: MCF-7 cells	81% DLE, 49% DRE	This system displayed dual pH/NIR-responsive drug release. The GNRs act as a probe for PTT.	[109]
	CNH/DCA-HPCHS	in vitro: 4T1 cells; in vivo: mice bearing 4T1 tumours		This system combines chemotherapy with NIR-mediated PTT for the treatment of cancer in vitro and in vivo. The nanocarrier displays pH-dependent drug release; however, no DOX quantification for drug loading/release was included.	[110]
	oxiCNH/SA-mAb	in vitro: MCF-7 and HEK293 cells; in vivo: CR male mice bearing subcutaneous hepatic H22 tumours	100% DLC, 67% DRE, with pH-dependent drug release	DOX is released in the endosomes of MCF-7 cells. Specific targeting of VEGF-containing cancer cells over healthy HEK293 cells was achieved by using mAb as a targeting ligand. This nanocarrier was more effective than free DOX both in vivo and in vitro, whilst showing reduced liver and cardiac toxicity.	[9]
	PEG-SWCNT	in vivo: SCID mice bearing Raji lymphoma xenografts	100% DRE, with pH-dependent drug release	Increased tumour inhibition and reduced systemic toxicity compared to free DOX was determined through in vivo experiments.	[111]
	C60 fullerene	in vitro: CCRF-CEM, Jurkat, THP1, and Molt-16 cells		Long-term nanocarrier stability was observed in physiological saline solution. Complexation with C_60_ fullerene promoted Dox entry into leukemic cells, resulting in ≤ 3.5 higher cytotoxicity compared to free DOX.	[112]
	Fullerenol	in vitro: MCF-7 and MDA-MB-231 cells; in vivo: zebrafish embryo (*Danio rerio*, wild type)		This formulation resulted in enhanced uptake, decreased proliferation, and remarkable cytotoxicity of DOX in breast cancer cells. A decrease in zebrafish embryotoxicity compared to DOX alone was noted. No drug loading or release analysis was included in this study.	[113]
	C82 fullerene-cRGD	in vitro: NCl-H2135 cells	Redox-dependent DOX release, triggered by increasing glutathione concentration	Significant cytotoxicity at low doses was observed due to enhanced cellular uptake relative to free DOX.	[114]
	DSPE-PEG-ND	in vitro: 4T1 cells; in vivo: BALB/c mice injected with 4T1 cells and SD rats	Approximately 95% DLE and 34% DRE	Favourable circulation time, accumulation in tumour tissues, and ability to deliver DOX to tumour cell nucleus was observed. This leads to a significant enhancement of DOX efficacy and biocompatibility.	[115]
	ND-PEG	in vitro: A549 cells	65% DLC, approximately 60% DRE	The nanocarrier was readily soluble in PBS and water, and unloaded ND-PEG displayed excellent biocompatibility. This system was able to deliver DOX directly to cancer cells.	[116]
	Carbon nanoring (CNR)	in vitro: A549, Hela, L929, and BEAS-2B cells	51% DLE, approximately 80% DRE, with pH-mediated drug release	Enhanced antitumor activity compared to free DOX was achieved. The DOX-CNR system was highly selective, with much higher cytotoxicity for cancerous cells than normal cells. DOX was released in the nuclei of cancer cells.	[117]
	GA-MWCNTMA-MWCNT	in vitro: MDA-MB-231 and MCF-7 cells	97% DLE and 71% DRE for MA-MWCNTs, and 96% DLE and 72% DRE for GA-MWCNTs, with pH-controlled drug release	The unloaded nanocarrier displayed high biocompatibility, whilst the drug-loaded systems showed slightly higher cytotoxicity than DOX in cancerous cells. Overall, the systems could effectively target cancer cells and deliver DOX to them.	[118]
	SWCNT-PEG-PEI-FA-CS	in vitro: MCF-7 cells	74% DLE and approximately 60% DRE, with pH-mediated drug release	Excellent dispersibility, cellular uptake, and antitumor activity was observed in this system. This nanocarrier caused apoptosis in cancer cells by triggering ROS overproduction.	[6]
	Fucoidan-decorated silica–carbon nano-onion (FSCNO)-HM	in vitro: HUVEC, NCI/ADR-RES, A2780ADR, and OVCAR-8 cells; in vivo: NU/NU nude mice xenografted with NCI/ADR-RES, A2780ADR, and OVCAR-8 cells	4% DLC and approximately 60% DRE for DOX, with NIR-triggered drug release	The co-delivery of DOX and P-gp pump inhibitor (HM) to counteract chemotherapy resistance increased DOX bioavailability and cytotoxicity. The nanosystem could effectively target and treat both drug- and non-drug-resistant tumour models with decreased side effects. FSCNOs also display photothermal capabilities.	[119]
	ATRA-ND	in vitro: HepG2, MCF-7, and CRL1730; in vivo: HepG2- and MCF-7-induced tumour-bearing BALB/c nude mice	25% DLC and >80% DRE, with pH-triggered drug release	Co-delivery of DOX and ATRA in conjunction with ultrasound treatment. ATRA enhances DOX cytotoxicity, whilst ultrasound enhances nanocarrier permeability into tumour blood vessels. This approach resulted in effective DOX delivery and dramatic tumour growth inhibition.	[120]
	ND-PEG-HYD-FA	in vitro: HeLa, HepG2, MCF-7, and CHO cells; in vivo: BALB/c mice inoculated with HepG2 cells	8% DLC, 85% DRE, with pH-triggered drug release	This nanocarrier has fluorescence imaging capabilities. Low drug release at pH 7 was due to the use of a cleavable hydrazone linkage. The FA-targeting ligand facilitated endocytosis and the rapid build-up of nanocarrier in cells. In vivo, experiments showed better tumour inhibition along with minimal cardiotoxicity, hepatotoxicity and nephrotoxicity when compared to free DOX.	[121]
	CNT/HA-DMPE	in vitro: MDA-MB-231 and A2780 cells	20% DLC, 18% DRE, with pH-triggered drug release	This biocompatible nanocarrier is highly stable in biological buffers. The system is easy to prepare and exhibits good targetability towards CD44-overexpressing cancer cells, resulting in a remarkable increase in DOX efficacy.	[22]
	AL-PEG-MWCNTs	in vitro: OPM2, HOS, MCF-7, 3T3-L1, and Raji Burritt’s lymphoma cells; in vivo: CD-1, Ath/nu, and CB.17 SCID mice; ex vivo: C57BL/6J bones	35% DLC, 51% DRE	This system uses individual MWCNTs to improve DOX pharmacokinetics. Its biocompatibility was confirmed through neoplastic transformation, chromosomal aberration, and cytotoxicity assays. Treatment-related weight loss was eliminated in a lymphoma in vivo model, whilst retaining DOX efficacy.	[122]
	Alginate–urea CDs	in vitro: MFC-7 cells	>70% DLE, ~45% DRE, with pH-triggered drug release	This fluorescent CD system is nontoxic and could be incorporated into hydrogels as a toughening agent. The nanohybrid displayed controlled drug release.	[123]
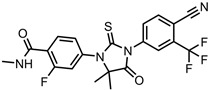 Enzalutamide—androgenreceptor antagonist	TP-PEG-Aminated GQD	in vitro: C4-2B and LNCaP cells; in vivo: BALB/c nude mice bearing C4-2B tumours	60% DLE, 95% DRE, with redox-sensitive drug release	This nanocarrier was rapidly uptaken by prostate cancer cells via endocytosis. This caused the effective inhibition of said prostate cancer cell growth in vitro and enhanced targetability and reduced side effects compared to the free drug in vivo.	[124]
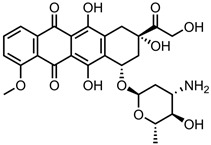 Epirubicin (EPI)—topoisomerase II inhibitor	SWCNT-DSPE-HA	in vitro: A549, AGS, and Taxol-resistant A549 cells	47% DLC, 60% DRE, with pH-mediated drug release	This system displayed high cancer cell targetability and biosafety. By facilitating the accumulation of EPI in cells by CD44-receptor-mediated endocytosis and preventing EPI efflux by P-gp, this system markedly improved the EPI anticancer efficacy in drug-resistant cancer cells.	[125]
Fe_3_O_4_-MWCNTs	in vitro: T24 and 5637 cells; in vivo: MNU-induced rat-bladder tumour model	40% DLE, 100% DRE, with magnetic-field-triggered EPI release	This system displayed prolonged retention, enhanced antitumor activity, and enhanced cytotoxicity compared to free EPI. Low systemic toxicity was also seen in vivo. The nanocarrier was also simple and quick to make.	[126]
	ND	in vitro: LT2-MYC cells; in vivo: FVB/N mice with MYC-induced tumours	48% DLE, >80% DRE, with pH- and intracellular charged protein-triggered release	This nanocarrier prevents the efflux of EPI by ABC transporters to counter chemoresistance in cancer stem cells.	[127]
	CD-TR-TM	in vitro: SJGBM2, CHLA266, CHLA200, and U87 cells	75% DRE with no information provided on drug release	No reference study was performed with healthy cells. The synergistic anticancer effect of the EPI/TM combination, combined with TR-facilitated targeting, drastically reduces cancer cell viability, even at low concentrations.	[128]
	HPPH/CPP-PEG-GO	in vitro: MG-63 cells; in vivo: osteosarcoma xenograft nude mouse model	Approximately 70% DRE, with no quantitative drug loading information provided	The synergistic effect of HPPH-mediated PDT and EPI chemotherapy allows for control over cancer cell growth. The incorporation of CPP further increases nanocarrier effectiveness by improving cancer cell targeting and internalisation.	[129]
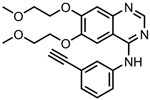 Erlotinib—tyrosine kinaseinhibitor	GO-PEG	in vitro: NPC TW01 cells	38% DLC, 98% DRE, with pH-mediated drug release	This nanoformulation achieved suppression of nasopharyngeal cancer migration, proliferation, and invasion via several molecular mechanisms.	[130]
Carboxylated NDs	in vitro: A549, NCI-H460, and NCI-H1975 cells	~57% DLE, no quantitative drug release data shown	This nanocarrier caused a decrease in drug-resistant cancer cell viability. Efficient uptake of nanocarrier via clathrin-dependent endocytosis was the key to its effectiveness, and the system was preferentially consumed by cancer cells.	[131]
	GO	in vivo: mice	30% DLC, 80% DRE, with pH-triggered drug release	The drug was released in a quick-burst fashion.	[132]
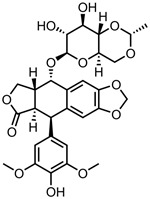 Etoposide (Et)—topoisomerase II inhibitor	GO-COOH	in vitro: HepG2 and RPMI-1640 cells	83% DLE, 98% DRE	The nanocarrier improved the cytotoxic effect of Et, with no change to its apoptosis pathway. The sustained release of Et offered by this formulation allowed for higher cytotoxicity and efficiency compared to the free drug.	[133]
oxiCNH/PEG-PA	in vitro: A549 and A549R cells; in vivo: BALB/c nude mice inoculated with A549R cells	39% DLC, 81% DRE, with NIR-triggered Et release	Slow and steady delivery of Et was observed at neutral pH, which was accelerated 1.5× upon NIR irradiation. CNHs are also photosensitizers, and the synergistic effect of PTT and Et chemotherapy killed multidrug-resistant cells by combating P-gp-mediated drug efflux.	[134]
	FA-CβCDT-MSCD	in vitro: HeLa and HepG2 cells	14% DLC, 25% DRE, with pH-mediated drug release	This nanocarrier displayed preferential targeting of folate-receptor-overexpressing cells. The encapsulation of Et in cyclodextrin prevented premature drug release.	[135]
	oxiMWCNT-PEG-Aso	in vitro: DMS53 and NCIH2135 cells	45% DLC, 88% DRE, with pH-sensitive drug release	This biocompatible nanocarrier allowed for the cellular internalisation of negatively charged nucleic acids, such as Aso. Aso binding increased the chemosensitivity of drug-resistant lung cancer cells, leading to superior cytostatic efficacy compared to the free drug. This system also displayed good aqueous dispersibility and low haemolytic activity.	[136]
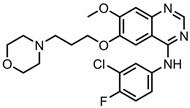 Gefitinib (GEF)—tyrosinekinase inhibitor	GO-PVP	in vitro: PA-1 and IOSE-364 cells	46% DLE, 35% DRE, with GEF release at neutral pH	This biocompatible nanocarrier is a combination therapy with quercetin and was found to enter cells via receptor-mediated endocytosis. The synergistic effect of GEF and quercetin results in significant therapeutic efficacy with ovarian cancer cells, higher than that of drugs delivered separately.	[137]
PEG-CQD-PVA-PLA	in vitro: NCI–H522 cells	~65% DRE, with no quantitative drug loading data shown	The PLA microspheres degrade via a hydrolytic reaction in acidic conditions, releasing GEF. The nanosystem delivered the drug directly to lung cancer cells, resulting in a significant decrease in the IC50 value compared to free GEF.	[138]
	GO nanosheets		43% DLC, 51% DRE, with pH-triggered drug release	Effective control over GO nanosheet size was achieved; GEF was converted to nanocrystals and then loaded onto GO.	[139]
	GO-HA	in vitro: A549 and HELF cells; in vivo: BALB/c nude mice bearing A549 tumours	13% DLC, 60% DRE, with redox-dependent GET release (glutathione-mediated)	This nanocarrier displayed efficient cancer cell uptake via CD44-receptor-mediated endocytosis, resulting in a significant enhancement of the GEF efficacy. The system showed stronger tumour inhibition than the free drug in vivo, with no obvious side effects.	[140]
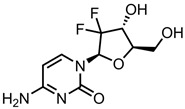 Gemcitabine (GEM)—ribonucleotide reductaseinhibitor	HA-PEG-MWCNT	in vitro: HT-29 cells; in vivo: SD rats bearing HT-29 tumours	90% DLE, ~85% DRE, with a pH-mediated, sustained drug release profile	A reduction in haemolytic activity and remarkable improvement in pharmacokinetic parameters compared to free GEM was observed. This was due to improved cellular internalisation, leading to enhanced anticancer effects.	[141]
SWCNT-PEG	in vitro: A549 and MIA PaCa-2 cells; in vivo: B6 athymic nude mice (both healthy and with A549 tumours)	37% DLC, ~80% DRE, with pH-triggered drug release, via a hydrolysable ester bond	This nanocarrier system accumulates in tumour cells and releases considerable amounts of GEM, resulting in the inhibition of tumour growth and a reduction in GEM side effects. This system also improved the stability of GEM.	[142]
	PEG-Fe3O4@GO@mSiO2-FA	in vitro: A431 cells	14% DLC, 85% DRE, with pH-triggered drug release	This system demonstrated enhanced GEM cytotoxicity and cellular uptake.	[143]
	ND-PEI-PAA-PEG-GFLG	in vitro: BxPC-3; in vivo: BALB/c nude mice xenografted with BxPC-3 tumours	No quantitative drug loading/release studies were performed	Significant nanocarrier stability was observed in physiological conditions, with long-term circulation due to PEG attachment and enzyme-sensitive GEM release. This system showed similar anticancer effects in vitro and a significant increase in antitumour effects in vivo compared to free GEM.	[8]
	ND-PEG	in vitro: AsPC-1 cells	No quantitative drug loading/release data were provided	The fluorescent NDs provide imaging and cell-tracking capabilities, and whilst no cytotoxicity enhancement compared to free GEM was seen, the nanocarrier successfully delivered GEM directly to pancreas cancer cells.	[144]
	ND@PHEA-co-POEGMEA	in vitro: AsPC-1 cells	7% DLC, ~100% DRE, with pH-triggered drug release	GEM was incorporated into HEA polymer and then loaded onto NDs. This resulted in slow, sustained GEM release delivered directly to cancer cells, with a similar IC_50_ value to free GEM.	[145]
	GO/MMT/CS	in vitro: MDA-MB-231 cells	99% DRE, with no qualitative drug loading data shown and pH-triggered drug release	GEM intercalated between MMT silicate layers, preventing burst release. The unloaded nanocarrier is nontoxic, and the sustained release of GEM from the system results in the excellent growth inhibition of breast cancer cells.	[146]
	FA-CS/Fe_3_O_4_/GO		22% DLC, 83% DRE, with pH-triggered drug release	This system was tested in simulated cancer fluid and simulated human blood, and it is also responsive to external magnetic fields.	[147]
	rGO	in vitro: A549, HEL-299, and NIH-3 T3 cells; in vivo: BALB/c mice (both healthy and xenografted with A549 tumours)	No quantitative drug loading/release data shown. NIR-triggered drug release	This system displayed excellent in vitro cytotoxicity against multiple lung cancer cell lines and low systemic toxicity, and a significant enhancement in antitumour activity compared to the free drug in vivo.	[148]
	CD	in vitro: MCF-7 and HeLa cells	34% DLC, 54% DRE, with pH-mediated drug release	This highly fluorescent nanosystem increased the potency of GEM with minimal cytotoxic effects on healthy cells. This was due to the effective transport and delivery of GEM to cancer cells.	[149]
	MWCNT-LE	in vitro: MCf-7 cells; in vivo: BALB/c nude mice bearing MCf-7 tumours	Approx. 31% DLE, with no qualitative drug release data	This combination chemotherapy and PTT nanocarrier achieved good antitumour activity in vitro and in vivo, with reduced side effects seen in animal studies.	[150]
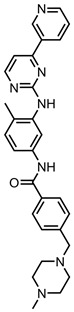 Imatinib—tyrosine kinaseinhibitor	N-prGO-CMC		74% DLC, 58% DRE, with pH-triggered IM release	The drug is bound to the nanocarrier via π–π stacking and hydrogen-bonding interactions.	[151]
ZnO/CNT@Fe_3_O_4_	in vitro: CML-derived K562 cells	No quantitative drug loading/release data shown	The system causes a significant decrease in leukaemia cell viability and metabolism. Cell death was caused by apoptosis via an ROS-dependent mechanism.	[152]
GQD	in vitro: RPMI 8226, MDA-MB-231, and NCI-ADR/RES cells	No quantitative drug loading/release data given	The system demonstrated efficient internalisation and remarkable cytotoxicity for cancer cells and was shown to localise in nuclei of neoplastic cells.	[153]
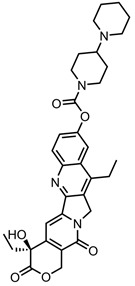 Irinotecan—topoisomerase I inhibitor	Fe_3_O_4_-GO-CS-PEG		3% DLC, 90% DRE, with pH-triggered drug release	A large percentage of drug bound to this nanocarrier was released at neutral pH. This system has magnetic targeting capabilities.	[154]
CD-PEG-BT	in vitro: MDA-MB231 and MCF-7 cells	23% DLC, 90% DRE, with NIR-triggered drug release	This combination chemotherapy/PTT nanosystem caused drug-resistant breast cancer cell death via necrosis and apoptosis pathways. The CDs also have fluorescence imaging capabilities.	[155]
Fe_3_O_4_-GO-CS-UA-GRP-SLP2	in vitro: U87 cells; in vivo: BALB/c nude mice with U87 tumours	58% DLC, 62% DRE, with pH-responsive drug release	This system displayed excellent targeted drug delivery, antitumour efficacy, and prolonged animal survival in brain tumour models using intravenous administration coupled with magnetic guidance. A 4.9× increase in drug uptake compared to the free drug was measured in vitro. A 6.5× enhancement in the ability to cross the blood–brain barrier compared to the free drug was seen in vivo. Highly biocompatibility was also seen in vivo.	[156]
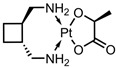 Lobaplatin—DNA alkylating agent	E2-PEG-CNT	in vitro: MCF-7 cells; in vivo: healthy C57BL/6 mice	No qualitative drug loading/release data shown	This nanocarrier has sustained release properties, with no obvious side effects and an increase in retention time compared to the free drug in vivo. The effective killing of cancer cells was seen in vitro due to E2-mediated targeting.	[157]
	PEG-CNT-FITC	in vitro: HepG2 cells	72% DLC, 80% DRE, with pH-triggered, sustained drug release and fluorescence imaging properties	This nanocarrier can effectively enter and kill liver cancer cells, with increased cytotoxicity observed up to 72 h.	[158]
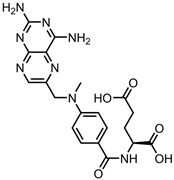 Methotrexate (METX)—nucleotide synthesis inhibitor	oxiSWNH-PEG-Tf	in vitro: MAD-MB-231 and HepG2 cells; in vivo: ICR mice carrying H22 tumours	15% DLC, 55% DRE; this system utilises pH-triggered drug release	The drug is released quickly at neutral pH, which could cause toxicity due to premature leakage. This system displayed favourable tumour targeting, cytotoxicity, and accumulation.	[25]
CMC-GO	in vitro: NIH-3T3 and HT-29 cells; in vivo: BALB/c mice and nude mice xenografted with HT-29 tumours	39% DLC, 82% DRE, with pH-triggered drug release	This system reduced drug toxicity against healthy cells and facilitated a higher plasma concentration, superior tumour cytotoxicity, and liver cancer metastasis inhibition compared to free METX.	[159]
Hydroxylated C_60_ fullerene	in vitro: MDA-MB-231 cells; in vivo: Wistar rats	No quantitative drug loading data shown, with 85% DRE and pH-sensitive drug release	This nanosystem drastically increased plasma half-life and AUC compared to the free drug, resulting in a large reduction in its IC_50_ value. Enhanced bioavailability, erythrocyte compatibility, protein binding, and haemolysis in human blood compared to free METX were also observed.	[23]
	AF-FA-^99m^Tc-MWCNT	in vitro: A549 and MCF 7 cells; in vivo: New Zealand rabbits and FR+ EAT-bearing mice	33% DLC, >85% DRE, with pH-triggered drug release achieved via a cleavable ester linkage	Effective targeting and treatment of folate-receptor-overexpressing cancer cells with reduced side effects and increased efficacy in vivo was observed. This nanocarrier also had fluorescence imaging and radio-tracing capabilities.	[160]
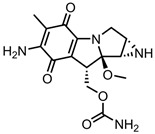 Mitomycin C (MMC)—DNA alkylating agent	TAT-graphene	in vitro: OCM-1 and ARPE-19 cells	22% DLC, 45% DRE, with pH-triggered drug release; however, the release in acidic and neutral environments was very similar	This system could specifically target cancer cells over healthy cells in a co-culture environment. The nanocarrier localised in the cancer cell nuclei, resulting in strong growth suppression.	[161]
CD	in vitro: MCF-7 cells	Approximately 80% DRE, with no quantitative drug loading information provided and pH-mediated MMC release	MMC was bound to the CDs via hydrogen bonding. This nanocarrier showed high affinity towards cancer cell membranes and could effectively enter them and accumulate. This resulted in a significant improvement in anticancer potency compared to free MMC.	[162]
	SWCNT-PEG-CWKG(KWKG)_6_	in vitro: A549 cells	Approximately 80% DRE, with pH-mediated MMC release	The unloaded nanocarrier showed high biocompatibility whilst the drug-loaded system exhibited similar anticancer efficacy to free MMC.	[163]
	Graphene-BODIPY-PEG	in vitro: HeLa cells	10% DLC, with no quantitative drug release data given	This nanocarrier possesses excellent photothermal conversion efficiency and ROS production capabilities for combination PTT/PDT. The system also has fluorescence and photothermal imaging capabilities and displayed outstanding anticancer effects.	[14]
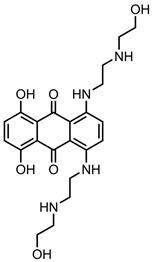 Mitoxantrone (MTX)—topoisomerase II inhibitor	GO-Fe_2_O_3_-MitP	in vitro: A549 cells	19% DLC, 38% DRE, with magnetic-field-triggered MTX release	MitP grafting improves the drug loading capability of this nanocarrier. Successful targeting and disruption of tumour mitochondria was achieved, causing cell death.	[164]
oxiMWCNT	in vitro: NIH3T3 and MDA 231 cells	95% DLC, 30% DRE	MTX was bound to oxiMWCNTs via electrostatic interactions. This formulation resulted in increased MTX efficacy; however, the system was more toxic to healthy cells than cancerous cells.	[165]
ND	in vitro: MDA-MB-231 and MDA-MB231-ABCG2 cells	87% DLE, 80% DRE, with pH- and soluble-protein-triggered MTX release	MTX release was found to be higher in FBS than in water, suggesting that the presence of soluble biological matter increases the DRE. A marked increase in MTX retention and efficacy was observed when using this nanocarrier.	[166]
	oxiSWCNT-PEG-FA	in vitro: HeLa cells	~35% DLE, 55% DRE, with pH-mediated, sustained drug release	This system selectively targeted cancer cells.	[167]
	EXO-GO-CO-γPGA	in vitro: MDA-MB-231 and BEAS-2B cells	73% DLE, 56% DRE, with pH-mediated, sustained drug release	This nanocarrier displays excellent cancer cell targetability. The attachment of exosomes was found to improve drug loading, pH response, and biocompatibility.	[168]
	rGO-PEG-SB	in vitro: 4T1, CT26, and bone marrow macrophages + DCs harvested from BALB/c mice; in vivo: BALB/c mice with 4T1 tumours	48% DLE, with no quantitative drug release data provided and NIR-triggered drug release	The synergistic combination of PTT, chemotherapy, and immunotherapy facilitated the destruction of local primary tumours and distant metastases in an in vivo model. rGO acted as a photosensitizer, whilst the SB immunotherapeutic increased effectiveness of rGO and MTX by TGF-β inhibition.	[169]
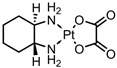 Oxaliplatin (OP)—DNA alkylating agent	GO-PNVCL-PGA	in vitro: MCF-7 cells	12% DLC, 80% DRE, with pH- and thermal-responsive OP release	Improved cytotoxicity compared to free OP against breast cancer cells was observed. The blank nanocarrier is nontoxic.	[170]
GO-HSA NPs	in vitro: HFFF2	61% DLE, ~97% DRE, with pH-mediated, sustained drug release	The use of HSA nanoparticles increased nanocarrier biocompatibility.	[171]
	MWCNT-PEG	in vitro: HT29 cells	43% DLC, no quantitative drug release data	A drastic increase in cytotoxicity towards human bowel cancer cells was observed.	[172]
	GO-CB [7]/Ce6/AQ4N/ADA-HA	in vitro: B16 and L02 cells; in vivo: C57BL/6 mice (both healthy and carrying B16 tumours)	58% DRE, with spermine-triggered OX release	Significant antitumour efficacy was observed, resulting from synergistic PTT (GO)/PDT (Ce6), and chemotherapy (OP and AQ4N) properties. The enhanced hypoxia resulting from PTT bolsters the effects of chemotherapy drugs. This strategy has the benefit of being able to noncovalently attach nonaromatic drug molecules via host–guest complex formation.	[173]
	TAT-BT-PEI-MWCNT	in vitro: C6 glioma (cell and tumour spheroid) and CHEM-5 and L02 cells; in vivo: mice bearing C6 tumours	19% DLC, with no quantitative drug release data shown	The nanocarrier shows enhanced blood–brain barrier penetration compared to free OX, resulting in a significant decrease in the IC_50_ value. The system shows low cytotoxicity towards healthy cells; however, a build-up of cerebrospinal fluid was noticed during treatment of in vivo models.	[174]
	GO-CS-FA	in vitro: LX-2 and SKOV3 cells	34% DLC, ~80% DRE	This nanoformulation shows similar potency to free OX in ovarian cancer cells and good biocompatibility.	[48]
	CD	in vitro: L929, HeLa, and HepG2 cells	4% DLC, with redox-sensitive drug release	The CDs have multicoloured emission capabilities and high fluorescence stability. This system shows good biocompatibility, bio-imaging, and anticancer effects both in vitro and in vivo.	[175]
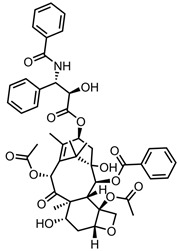 Paclitaxel (PTX)—microtubule growth inhibitor	GO-MA/FA	in vitro: MDA-MB-231 cells; in vivo: SD rats with DMBA-induced mammary carcinoma	40% DLC, 71% DRE	PTX was attached to GO via π–π stacking and hydrophobic interactions. This system inhibited cancer cell growth in vitro and reduced tumour size in vivo via cell cycle arrest and apoptosis. Highly specific targeting and drug release for folate-receptor-overexpressing cancers was observed.	[176]
SWCNT/DOA-PEG-FA	in vitro: MCF-7 cells; in vivo: athymic nude mice with MFF-7-induced tumours and BALB/c mice	No quantitative drug loading or release data were provided	This system displayed high specificity, biocompatibility, and efficacy compared to free PTX in vitro. In vivo, studies revealed significant tumour growth inhibition, with no side effects on the blood and major organs of mice observed.	[177]
	PLA composite nanofibers/C_70_ fullerene	in vitro: HepG2 cells	No quantitative drug loading information, 72% DRE	Control of in vitro PTX release profile was achieved by varying fullerene content. Successful control of cancer cell growth was achieved using this nanoformulation.	[178]
	Rf-MWCNTs	in vitro: MCF-7 cells; in vivo: SD rats	82% DLE, 99% DRE	Haemolysis in human blood was much lower than free PTX. This system also showed better cytotoxicity than free PTX, with low systemic toxicity, favourable biodistribution, and renal excretion observed in vivo.	[179]
	Au–N-doped carbon nanotube cup (NCNC)	in vitro: MDSC cells; in vivo: B16 melanoma cells inoculated into C57BL/6 mice	36% DLE, with no quantitative drug release data shown	The PTX-containing NCNCs capped with Au nanoparticles exhibited strong surface-enhanced Raman scattering effects, allowing for extremely sensitive detection. A single injection of nanocarrier solution given to an in vivo model significantly reduced tumour growth and eliminated tumours in ~30% of mice. This system targeted lymphoid tissues surrounding tumours to boost the host immune system response.	[180]
	Graphene-PLA-PEG	in vitro: U-138 cells; in vivo: athymic nude Foxn1^nu^ mice implanted with U-138 tumours	4% DLC, 6% DRE	This nanocarrier is nontoxic and expressed long-term sustained drug release over 19 days. The system was found to be six times more potent than free PTX in vitro and is capable of reducing U-138 cell viability despite low drug loading and release.	[181]
	HMCS	in vitro: CNE cells; in vivo: nude mice with CNE tumours	19% DLC, no release experiments were carried out with PTX	HMCSs themselves exhibited strong antitumour effects through PTT.	[86]
	CNT-PMAA self-assembled micelles	in vitro: L929 and HeLa cells	36% DLE, 74% DRE, with pH-triggered drug release	The blank nanocarrier is nontoxic, with low haemolysis observed. Higher anticancer activity than free PTX was also noted. The self-assembly of CNT-PMAA is pH-dependant, and at low pH, the nanocarrier disassembles.	[182]
	FA-CD-GOx	in vitro: MDA-MB-468 (cells and tumour spheroids); TNBC and HEK293 cells	82% DLE, 95% DRE, with GOx and PTX release occurring at pH 7.4	GOx-induced cancer starvation, which had a synergistic effect with PTX chemotherapy, and resulted in significant cancer cell death. This biocompatible nanocarrier also efficiently targets cancer cells over normal cells.	[13]
	FCD_b_	in vitro: NIH3T3 and B16F10 cells	82% DLC, with no quantitative release data given	This nanocarrier targeted cancer cells in a co-culture with healthy cells due to its biotin-targeting ligand. It caused the selective sensing and activation of H_2_O_2_, and it also has fluorescence imaging capabilities.	[183]
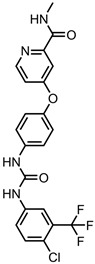 Sorafenib—tyrosine kinaseinhibitor	rGO	in vitro: SGC7901 cells	No quantitative drug loading/release data	A significant increase in cytotoxicity compared to the free drug in gastric cancer cells was observed, with apoptosis being the main mechanism of cell death.	[184]
CS nanoparticles-FA	in vitro: HepG2, HDFa, and HT29 cells	19% DLC, 91% DRE, with pH-triggered, sustained drug release profile	This system showed negligible toxicity to healthy cells and enhanced anticancer properties compared to the free drug.	[185]
CNT-PEG	in vitro: HepG2 cells; in vivo: Wistar rats, both healthy and with DENA-induced liver tumours	95% DLC, ~100% DRE	A negligible change in potency or morphology was observed over a 3-month stability study. This system displayed superior anticancer abilities compared to the free drug both in vitro and in vivo.	[186]
DSPE-PEG-ND	in vivo: healthy rats and BALB/c mice carrying BGC-823 tumours	~90% DRE, with no quantitative drug loading data shown	A 14-fold increase in drug concentration in tumour tissues compared to the free drug was observed. This caused considerable growth inhibition for the in vivo gastric cancer model. A 7.64× increase in oral bioavailability compared to the free drug was also seen in vivo.	[187]
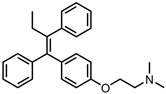 Tamoxifen (TAM)—oestrogen receptor antagonist	rGO	in vitro: MCF-7 cells; in vivo: BALB/c mice bearing MCF-7 tumours		This system displayed lower cytotoxicity than free TAM but had promising photothermal properties. No quantitative drug loading/release data were shown. The nanocarrier has excellent stability.	[188]
NGR-SWCNT-PF68	in vitro: 4T1 cells; in vivo: BALB/c mice carrying 4T1 tumours		This targeted nanocarrier could effectively enter cancer cells whilst retaining TAM cytotoxicity. Receptor-mediated tumour targeting combined with PTT capabilities resulted in significant anticancer efficacy in vivo. No quantitative drug loading/release data shown.	[189]
	MWCNT-LE	in vitro: MCF-7 cells	28% DLC, no quantitative drug release data	Enhanced cellular uptake, apoptosis, and antitumour activity (compared to free TAM) were observed. This system also has PTT properties. LEN acts as both a dispersant and a potent anticancer agent itself. Combination chemo–PTT results in cancer cell destruction at low drug concentrations.	[190]
	C_60_ fullerene-Gly	in vitro: MCF-7 cells; in vivo: Wistar rats	66% DLC, 85% DRE, with pH-triggered TAM release	This nanocarrier vastly improved the pharmacokinetic properties and cytotoxicity of TAM against breast cancer cells. The system displayed minimal haemolysis towards human blood. The bioavailability, half-life, and cancer cell penetration of the drug were all markedly improved.	[191]
	TEG-MWCNT-quercetin	in vitro: MDA-MB-231 cells; in vivo: Wistar rats	No quantitative drug loading info, 93% DRE, and pH-triggered drug release using a TAM prodrug	This haem-compatible formulation reduced the IC_50_ values and increased the uptake in drug-resistant cells. These favourable properties carried over to in vivo studies, resulting in enhanced efficacy, pharmacokinetics, and biocompatibility.	[192]
	DES-graphene	in vitro: MCF-7 cells	No quantitative drug loading/release data	Nanocarrier possesses acute, selective anticancer activity achieved through intracellular ROS-production-triggering cell cycle arrest.	[193]
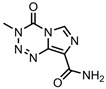 Temozolomide—DNAalkylating agent	GO-Fe_3_O_4_	in vitro: rat glioma C6 cells	90% DLC and 74% DRE, with pH-mediated drug release	The blank nanocarrier is biocompatible, whilst the loaded system showed better inhibitory effects than the free drug in rat glioma cells. This formulation also has strong magnetic properties.	[194]
rGO	in vitro: LN229 cells	84% DLC and 83% DRE, with electrochemically controlled drug release	This system retained the anticancer potency of temozolomide.	[195]
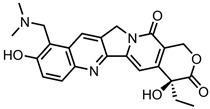 Topotecan—topisomerase Iinhibitor	GO/TT/CisP/cholesterol-DOX encapsulated in a DSPE-PEG nanocell	in vitro: HeLa cells	29% DLC and 40% DRE for TT, with pH-triggered drug release	This nanocarrier has fluorescence imaging capabilities and displays remarkably higher anticancer efficacy compared to the free-drug combination due to the synchronised targeting of multiple targets in cancer cells at once.	[196]
PEI-GO-TT-CisP	in vitro: HeLa cells	51% DLC for TT, no quantitative drug release data	This nanocarrier has subcellular-organelle-targeting capabilities and can effectively impair the mitochondria of cervical cancer cells, leading to cell death. This resulted in a 4.4× decrease in IC_50_ compared to the free-drug cocktail.	[197]
	TT/GO-CD/DOX	in vitro: HeLa cells	16% DLE and 77% DRE for TT, with sustained, pH-triggered drug release	This system demonstrated superior anticancer efficacy compared to free drugs and single-drug-loaded nanocarrier due to the synergistic effect between the drugs and GO.	[198]
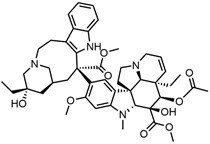 Vinblastine—mitosis inhibitor	CQD	in vitro: Hela, HGC-27, A549, MCF-7, CF-STTG, and Vero cells; in vivo: NOD-SCID mice carrying A549 tumours	95% DLC, with no qualitative drug release data	The cytotoxicity of vinblastine was reduced in normal cells and increased in cancer cells compared to the free drug. Significant inhibition of tumour growth was observed in vivo, with no liver toxicity. The synergistic combination of chemotherapy and PTT allowed for the control of cancer cell growth.	[199]
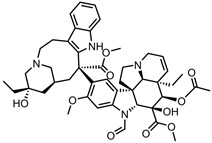 Vincristine—mitosis inhibitor	oxiSWCNH-PEG-mAb	in vitro: MCF-7 and HUVEC cells; in vivo: ICR mice carrying H22 tumours	35% DLC, 80% DRE, with pH-triggered, sustained drug release	Effective cancer cell killing in MCF-7 cells and treatment of liver cancer in mice with superior efficacy compared to the free drug. The system displayed reduced systemic toxicity compared to the free drug due to mAb-facilitated targeting.	[200]

**Table 2 pharmaceutics-15-01545-t002:** Index of anticancer drug molecules in the CNM nanocarrier database.

Drug	Page	Drug	Page
5-fluorouracil	5	epirubicin	23
6-mercaptopurine	6	erlotinib	24
anastrozole	7	etoposide	24
bortezomib	7	gefitinib	25
capecitabine	7	gemcitabine	26
carboplatin	7	imatinib	27
carmustine	9	irinotecan	28
chlorambucil	9	lobaplatin	29
cisplatin	9	methotrexate	29
curcumin	13	mitomycin C	30
cyclophosphamide	15	mitoxantrone	30
cytarabine	15	oxaliplatin	31
dabrafenib	16	paclitaxel	33
dasatinib	16	sorafenib	34
daunorubicin	16	tamoxifen	35
decitabine	17	temozolomide	36
docetaxel	17	topotecan	36
doxorubicin	19	vinblastine	37
enzalutamide	23	vincristine	37

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
