# Peer review of "Carbon Nanomaterials (CNMs) in Cancer Therapy: A Database of CNM-Based Nanocarrier Systems"

_pharmaceutics, 2023, doi:10.3390/pharmaceutics15051545_

Round 1

Reviewer 1 Report

The manuscript titled: "Carbon Nanomaterials (CNMs) in Cancer Therapy: A Database of CNM-based Nanocarrier Systems" reports an interesting and exhaustive database of 191 examples of carbon nanomaterials including graphene oxide, carbon nanotubes and carbon dots, demonstrating enhanced anticancer efficacy and reduced side effects.

In my opinion it could be considered for the publication after minor revision.

-) Typos should be corrected.

-) Carbon nanomaterials possess intrinsic photothermal properties mainly due to the nπ* state properly modulated by the doping elements (N, O, S,…..). A brief discussion about this property including the conversion yield efficiency definition, should be improve the better understanding the  CNMs photothermal effect.   Recent references focused on phototehrmal CNMs could be cited i.e.:

https://doi.org/10.1021/acsami.2c22500

https://doi.org/10.1016/j.colsurfb.2022.112628

Author Response

Dear Reviewer 1,

Thank you very much for your positive review, stimulating and insightful comments, which we believe have served to strengthen the quality of the manuscript. We have addressed all of your comments either as revisions to the manuscript or as comments, and we believe that in its present form, the manuscript is suitable for publication in Pharmaceutics. See below replies to the comments:

Comment 1: The manuscript has been extensively reviewed for spelling and grammar errors; these have been corrected.

Comment 2: A brief discussion on the nπ* state and the photothermal conversion efficiency of CNMs with up-to-date references has been included on page 40.

Reviewer 2 Report

The present work addresses a topic of importance for pharmaceutic and bio-medical research –  Carbon nanomaterials (CNMs) in cancer therapy: a database of CNM-based nanocarrier systems.

The abstract is well structured. In brief, it presents the main motivation for the current review –a comprehensive database of CNM-based nanocarrier systems incorporating approved chemotherapy drugs. It is based on an compiled from an extensive literature review.

The introduction is concise and correct. It summarizes the possible biomedical applications of CNMs, in the area of drug delivery and diagnostics, because of their unique and highly desirable physicochemical and mechanical properties. The authors comment on their size, biocompatibility, high tensile strength and ease of chemical functionalization. The authors critically analyze and discuss the database, suggesting possible directions for future research on CNM-based nanocarriers for anticancer drug delivery.

The aim could be more clear and focused.

Methods and all used metrics in the database construction are correctly described.

The main part of the review is well structured.

The presentation is detailed and logically developed. Particular attention is directed to systematizing in Table 2 the different CNM-based nanocarrier, chemotherapeutics, drug loading and release metrics, the used biological study models and experimental results, which makes the review helpful in the field of pharmaceuticals and biomedical research. The details are correct.

The discussion contains a critical analysis and summary of both the most established carbon nanomaterials and drug moieties. The synergistic effect of nanocarriers with toxic chemotherapeutics is also commented. This effect allows the administration of lower amounts of drugs to achieve the same therapeutic effect. CNT-based systems with high loading and release, as well as their biocompatibility and toxicity, are analyzed in order to overcome the obstacles to their application in clinical trials.

The review is based on 201 articles, the majority of them published during the last 5 years.

References are correct.

In conclusion, the manuscript's topic is very interesting and will stimulate the reader’s interest towards new and promising strategies for applying of CNM-based nanocarriers in cancer therapies. The review provides useful, summarized and systematized information. It is well-prepared and written. Finally, I strongly recommend the proposed review be accepted for publication in Pharmaceutics.

The quality of English is good. There are some typing mistakes, please check. 

Author Response

Dear Reviewer 2,

Thank you for your positive review, stimulating and insightful comments, which we believe have served to strengthen the quality of the manuscript. We have addressed all of your comments either as revisions to the manuscript or as comments, and we believe that in its present form, the manuscript is suitable for publication in Pharmaceutics. See below replies to the comments:

Comment 1: The abstract has been rewritten to focus more on the goals of the database itself. The end of the introduction (page 2) has been adjusted to make the aims of the main text of this review more clear.

Comment 2: The manuscript has been extensively reviewed for spelling and grammar errors; these have been corrected. 

Reviewer 3 Report

The manuscript entitled, ‘Carbon Nanomaterials (CNMs) in Cancer Therapy: A Database of CNM-based Nanocarrier Systems’ discussed carbon nanomaterials based nanocarrier systems. The article could be published after accounting the following comments;

1.      The abstract needs to be more focused. Why this review is different from others should be stated also.

2.      Several nanomaterials based on carbon are stated but one major nanomaterial, quantum is mission. As this is more recent, thus t should be included also. For this here are some article for your references: https://doi.org/10.1021/acsabm.2c00664; https://doi.org/10.1021/acsami.0c14527; https://doi.org/10.3390/cancers13091991; https://doi.org/10.3390/ijms222111783.

3.      It will be better if some mechanistic approaches of drug carrier/encapsulation are discussed.

Author Response

Dear Reviewer 3,

Thank you for your stimulating and insightful comments, which we believe have served to strengthen the quality of the manuscript. We have addressed all of your comments either as revisions to the manuscript or as comments, and we believe that in its present form, the manuscript is suitable for publication in Pharmaceutics. See below replies to the comments:

Comment 1: The abstract has been rewritten to focus more on the goals of the database itself. We have also included reasons why our review differs from other published works, for example, this article is the first comprehensive review of CNM-based anticancer systems that incorporates many different drug and CNM types.

Comment 2: The database contains entries on both carbon dots and carbon quantum dots, however, we have added more recent references to further highlight the benefits of these materials.

Comment 3: This review focuses on describing the composition, uses, and benefits of CNM-based nanocarrier systems, the mechanisms of drug encapsulation are outside the scope of this article.

Reviewer 4 Report

argeted delivery systems are one of the never-fading topics of modern chemistry and materials science. It is quite logical that such a convenient and popular platform as Carbon Nanomaterials will be used as a carrier.

Overall, the review is well written and detailed. Links to sources are relevant and cover a large time period. I would prefer to add individual carrier and individual drug to the IC-50 metrics, in the case of a two-component system. If they contain several substances and for them. This makes it possible to review the boele, and allows an assessment of the contribution of the platform to the toxic effect.

Carbon and carbon nanomaterials, due to their nature, are highly toxic, and this contribution must be accepted.

Some promising carbon materials, such as: carbon nanoribbons, oxidized modified carbon (amorphous), oxidized nanotubes, were not encountered in the work.

Author Response

Dear Reviewer 4,

Thank you for your positive review and for the stimulating and insightful comments, which we believe have served to strengthen the quality of the manuscript. We have addressed all of your comments either as revisions to the manuscript or as comments, and we believe that in its present form, the manuscript is suitable for publication in Pharmaceutics. See below replies to the comments:

Comment 1: Overall, the nanocarriers displayed excellent biocompatibility. In the case of IC50 values, only two examples of IC50 values of the nanocarrier alone were found (ref [152] [198]). In both cases, the nanocarrier was found to have no significant cytotoxicity and thus the decrease in cell viabilities observed was attributed to the anti-cancer drug alone. In the case where the nanocarrier was found to be cytotoxic, this was used as a justification for functionalization and the functionalized nanomaterial was then used for further investigation in the study. This makes comparisons between the IC50 value of the free drug(s) and that of the nanocarrier difficult.

Comment 2: Carbon nanomaterials have been found to display dose-dependent toxicity, and this has been addressed in the last paragraph of the review, where we discuss CNTs and fullerenes as examples and how this issue has been overcome in the literature.

Comment 3: Unfortunately, we could not find any suitable references for carbon nanoribbons of oxidised amorphous carbon. In the case of oxidised CNTs, numerous examples of these have been encountered in the literature and can be found in the database (see [44], [135], [164], [166] etc).

Round 2

Reviewer 3 Report

This can be published in its present form.